# Three-Dimensional Ultraviolet Fluorescence Imaging in Cultural Heritage: A Review of Applications in Multi-Material Artworks

**DOI:** 10.3390/jimaging11070245

**Published:** 2025-07-21

**Authors:** Luca Lanteri, Claudia Pelosi, Paola Pogliani

**Affiliations:** 1Department of Economics, Engineering, Society and Business Organization, University of Tuscia, Largo dell’Università, 01100 Viterbo, Italy; llanteri@unitus.it; 2Department for Innovation in Biological, Agro-Food and Forest Systems, University of Tuscia, Largo dell’Università, 01100 Viterbo, Italy; pogliani@unitus.it

**Keywords:** ultraviolet fluorescence imaging, cultural heritage, digital twins, 3D models, photogrammetry, sculptures

## Abstract

Ultraviolet-induced fluorescence (UVF) imaging represents a simple but powerful technique in cultural heritage studies. It is a nondestructive and non-invasive imaging technique which can supply useful and relevant information to define the state of conservation of an artifact. UVF imaging also helps to establish the value of an artwork by indicating inpainting, repaired areas, grouting, etc. In general, ultraviolet fluorescence imaging output takes the form of 2D photographs in the case of both paintings and sculptures. For this reason, a few years ago the idea of applying the photogrammetric method to create 3D digital twins under ultraviolet fluorescence was developed to address the requirements of restorers who need daily documentation tools for their work that are simple to use and can display the entire 3D object in a single file. This review explores recent applications of this innovative method of ultraviolet fluorescence imaging with reference to the wider literature on the UVF technique to make evident the practical importance of its application in cultural heritage.

## 1. Introduction

Ultraviolet-induced fluorescence (UVF) in the visible range is an imaging diagnostic tool. Along with other imaging techniques, UVF is widely used in the conservation and restoration of cultural heritage in order to obtain information about the surface state of conservation of artworks in a completely non-invasive and nondestructive way [1,2,3,4,5,6,7,8]. For this reason, it is almost imperative that UVF imaging be carried out before starting restoration, as this allows us to reveal the presence of inpainting, grouting, fluorescent materials, previously restored areas; in general, details not visible to the naked eye [9,10,11,12,13,14,15,16,17,18,19,20,21,22]. In Figure 1, four examples of UVF images are presented to highlight the great utility of UVF imaging in the study, interpretation and restoration phases of an artwork. In the case of the 18th-century tetramorphic wooden lectern shown in Figure 1A, UVF photography was particularly useful following cleaning of the surface artwork, as it enabled observation of the visible fluorescence emitted by the materials to be removed, i.e., the non-original layers that hid the original gilding. In the case of the large 18th-century wall paintings in Palazzo Nuzzi shown in Figure 1B, the UVF image, obtained by combining eighteen shots through a photogrammetric approach [21], allowed mapping of areas of non-original zinc white inpainting, which exhibits a lemon-yellow fluorescence. The UVF imaging of the 16th-century panel painting representing a Crucifixion shown in Figure 1C revealed very interesting detail of the pictorial construction in the face of St. John, in addition to highlighting a very poor state of conservation which required a restoration which has only recently been completed [22]. The face of St. John in fact exhibits a changed appearance in the UVF image, compared with the visible image, revealing a sophisticated construction technique which is not observed in the other faces of the panel painting [22]. Lastly, the case of the recently restored large canvas painting shown in Figure 1D, which was made at the end of the 19th century by the artist Pietro Vanni to faithfully reproduce the Renaissance wall paintings by Lorenzo da Viterbo in the Cappella Mazzatosta (Viterbo, Central Italy), provides an example of UVF utility in distinguishing painting materials apparently equal in the visible work. UVF imaging, in fact, allowed us to identify the use of two different whites in the flooring: lead white (no evident fluorescence); and zinc white (lemon-yellow fluorescence). This suggested that the artist probably painted the flooring at different times, using different painting pigments [12].

UVF imaging is a simple but powerful tool which enables restorers, art historians, archeologists, archivists and conservators to “look” at an artwork’s surfaces in a different way, that is, through the fluorescence response in the visible range induced by ultraviolet radiation [23,24,25,26]. The great relevance of the UVF technique to restoration is demonstrated in pioneering works by Rorimer [27], Eibner [28,29,30], Lyon [31] and other authors such as De la Rie, whose papers tell the story of UVF photography applied to the examination of artworks [9,10,11].

UVF photography has been widely used for paintings, sculptures, manuscripts, textiles, contemporary artworks, etc., and several papers in the literature describe how this technique has aided investigations of the surface states of artworks. However, only a very limited number of papers report the 3D photogrammetric application of UVF; further details of these will be provided later in this article.

Our search of the scientific literature was conducted using various keywords: 3D ultraviolet fluorescence imaging for cultural heritage, 3D multi-band imaging, 3D digital twins under ultraviolet fluorescence, ultraviolet fluorescence photogrammetry, ultraviolet 3D models, and 3D luminescence imaging. While, on the one hand, several papers were found concerning UVF imaging in 2D, on the other, only a very limited number of articles reporting 3D imaging under UVF were obtained [12,32,33,34,35,36,37,38,39,40,41,42,43].

Papers concerning topics related to 3D UVF imaging production, such as ultraviolet image acquisition, photogrammetry, digital model creation and applications of UVF imaging, are reported in the References section because they are relevant to the topic under discussion.

The first published papers proposing an innovative approach to creating 3D digital models under UVF were published in 2017 by Lanteri and Agresti [32], and by Lanteri et al. in 2018 [33] and 2019 [34]. In these papers, the authors reported for the first time a method to output UVF 3D models and to share them with end users, starting with photogrammetry and a Structure from Motion workflow, but with a variation in the acquisition procedure, as we will see later in the paper, in order to make the process simpler and more rapid. Subsequently, other authors presented the same approach for obtaining multispectral and UVF 3D models, including the authors of [35,36] and, more recently, the authors of [43].

It is worth noting that a 3D multi-band approach has also been employed for paintings (not for 3D artworks) to produce surface models at different wavelengths, but with a different methodology [44,45].

In the present review, our attention was focused on papers proposing the use of ultraviolet fluorescence imaging for obtaining 3D digital models of artworks in UVF as a powerful instrument for cultural heritage studies.

Detailed research has been conducted on the instrumental set-up used for obtaining ultraviolet fluorescence images by considering the UV sources, the filters and the cameras. The focus of this research has also been to underline the modalities of making digital twins usable for restorers and experts in the field, in order to make the fluorescence of the surfaces easily observable.

## 2. Methodology for Obtaining UVF 3D Digital Models: From the Acquisition Set-Up to the Tridimensional Model Creation and Sharing

The acquisition of photographic images of the fluorescence induced by ultraviolet radiation (UVF) in the visible range of the electromagnetic spectrum follows a procedure now standardized in the field of diagnostics for cultural heritage: the shots are taken in a darkroom using digital cameras equipped with a lens characterized by an intermediate focal length (most commonly 50 mm), in order to reduce by as much as possible the distortions and aberrations that affect wide-angle lenses. UV sources, filters and a tripod are also used [12,43,46]. The paper by Keats Webb is an interesting review of use of UV-induced visible luminescence for cultural-heritage documentation. Different aspects of techniques are reported concerning results obtained by applying UVL photography, with technical aspects of the methodology also highlighted [45]. Specifically, with respect to possible applications, the author illustrates various examples related to different kinds of artworks and materials such as pigments, binders, the components of varnish, paper, feather, glass, etc. [46].

UVF photography requires the use of a tripod to fix and stabilize the camera during acquisition. This is particularly relevant to ensure the camera’s stability during exposure times which are quite lengthy, usually around 15 s or more.

UV sources are the first essential requirement for obtaining UVF images. Different kinds of lamps may be used, such as Wood’s lamps, mercury-vapor lamps and xenon arc lamps, as well as the more recently introduced light-emitting diodes (LEDs) [5,15,38,47,48,49,50,51,52,53,54]. LED sources offer considerable advantages in terms of handling and the possibility of their being powered by rechargeable batteries, which means they can be operated in situ even when there is no power supply available [12,21,55]. Furthermore, LED lamps offer a controlled emission peak at 365 nm, ideal for UVF photography [43,47,49].

A panorama of possible UV lighting systems is supplied in two works available online [50,51]. However, the emission of the UV lamp is not always specified in papers [38]. Some authors report only the lamp typology, such as the long-arc-pressure mercury fluorescence tubes whose emission spectrum covers visible and infrared regions, necessitating the use of filters [5]. Other authors supply more specifications about UV lighting sources. These include a xenon arc lamp with long-wave UV light at 350 nm and 80 nm bandwidths which was used for documenting a damaged medieval manuscript and improving its readability [15]; and Xenon flashtubes with a bandpass filter at 350–400 nm associated with bandpass filters at 400–700 nm used to transmit visible radiation but block both UV and IR radiation [54].

Other authors report the use of low-pressure Hg tubes with an emission peak of approximately 365 nm coupled with blocking filters necessary for eliminating reflected UV radiation [48].

Relevant parts of the instrumental set-up also include the UV cut filters which are mounted in front of the photographic lens [12,56,57,58,59]. Sandwiches of UV-IR cut filters are often used to eliminate residual infrared and UV components that can affect the visible fluorescence response [12,47]. A wide range of filters used in UVF photography are reported by Crowther in an interesting recent paper [56].

Some papers report on the spectral characteristics of the used filters [12,43,51,56] although, in general, only typologies and cut-off limits are described. However, in all studies, the filters are used to acquire only the visible fluorescence induced by UV radiation, avoiding other spectral components that could affect the response of the materials.

One configuration recently used for UVF imaging is the combination of a UV-IR cut with another filter, such as those named A [12,47] and Schott BG40 [43], that seem to deliver optimal performance in acquiring only UV-induced visible fluorescence. A and Schott BG40 filters are near-infrared cut-off filters with a high steepness in this portion of the infrared region. Other authors propose the use of a UV-IR cut filter associated with a longpass filter having a cut-on wavelength of 420 nm, to reduce the small reflection of the blue component emitted by the UV LED sources [49].

Lastly, a digital camera (reflex or mirrorless), must be used for the capture of the images. Here, again, different kinds of apparatus may be used [12,51,60,61,62,63], including a low-cost compact digital camera [64].

Camera and lens typologies and settings are generally detailed in published papers; these include resolution, focal length, aperture, sensitivity (ISO) and numbers of acquired frames [36,37,38,43,47,65,66,67,68].

The most-used digital cameras have high resolution in the visible range. This is acquired when UV-induced fluorescence is emitted by surface materials.

As noted in the introduction, UVF photography is widely used for 2D technical documentation and as a diagnostic tool for paintings and 3D objects using a typical set-up, as reported by Cosentino [3]. However, the present review focuses on 3D applications of UVF imaging which are particularly useful for sculptures and tri-dimensional artworks in general.

The technique used does not differ from the Structure from Motion (SfM) workflow, which usually uses images taken in the visible range [68,69,70,71,72]. The basic principles of photogrammetry are still valid; in brief, a set of images must be acquired with a few important characteristics: 60–80% overlapping between frames, and limited optical and perspective distortions [73,74,75,76,77,78,79,80,81,82].

The workflow of the SfM process, which by its nature involves reconstructing a 3D digital structure starting from images taken by a moving sensor, involves the creation of a photographic studio set in a darkroom and very long acquisition times for UVF images. Due to the continuous movements of the camera, at least two UV lamps need to be placed around the object, in order to obtain complete coverage of the surface.

To reduce the acquisition times for the photographic set, it has been proposed by Lanteri [32,33] and subsequently by other authors [35,36,40,43] that the subject be positioned on a rotating support (360° around the vertical axis), and that the camera and lamps are kept in a fixed position. The acquisition of the frames and the management of the lighting is therefore much easier and faster, drastically reducing the time needed to obtain the set of photographic images.

In Figure 2 the screenshot of the software used for creating the model is displayed. The positions of the frames at the moment of the photographic shots are shown together with the newly rendered tri-dimensional model (at the center).

This procedure, in clear contrast with the principle of SfM, is not “perceived” during the image processing phase in the most-used commercial software [12].

In a nutshell, if the image matching algorithm recognizes, in the set of images, the object in the foreground in different positions, and the background with the same characteristics for the entire set, it aborts the calculation process (because it is based on a fixed object). Fortunately, in the case of UVF photographs, this is avoided. This is because image matching algorithms, when analyzing UVF images, process only the pixels that have captured the fluorescence of the irradiated surface of the object and are therefore recognizable on multiple images, but they do not distinguish the background pixels. In fact, the pixels that capture the background, which are generally very dark, are highly uniform, with confidence values so low that they are discarded by the image matching algorithm.

Regardless of the phenomenon described, most photogrammetry software is equipped with specific tools that allow the creation of cutout masks of individual images once they are loaded into the software. The masks allow us to exclude the background pixels from the calculation, concentrating the “image matching” process only on the pixels that represent the fluorescence produced from the object’s surface.

The digital product that is obtained at the end of the workflow is a 3D model characterized by photorealistic rendering and real dimensions, the latter being obtained by insertion of a scale bar into the scene or from knowledge of the real distance between two well-defined points of the model [12]. The model, although created with images of UV-induced fluorescence, returns a metrologically correct digital twin, i.e., a faithful 1:1 reproduction of the acquired artwork.

As highlighted in the literature, photogrammetry software used for the creation of 3D models starting from UVF images is available in both open-source and commercial packages.

As revealed by our bibliographic research, the commercial software generally used for creating tri-dimensional UVF models is Agisoft Metashape (version 1.8 and later) [83]. An available open-source alternative is AliceVision Meshroom [84].

Agisoft Metashape (version 1.8 and later) is stand-alone software that performs photogrammetric processing of digital images and generates 3D spatial data. The software allows for images from RGB or multispectral cameras to be processed, transforming them into spatial information in the form of dense point clouds, textured meshes and orthomosaics.

AliceVision Meshroom is free, open-source 3D reconstruction software based on the Alice Vision framework. The software has a good-quality node-based interface, which allows for the management of almost the entire workflow: photo alignment, sparse point cloud creation, high-quality mesh, and photorealistic texture generation. A limitation of this program is its inability to generate and display a dense point cloud within the program’s workspace. A dense point cloud can be exported; however, this requires additional work. Details concerning metric quality, spatial resolution and processing accuracy are not always reported in the literature on multispectral 3D imaging. In some cases, the authors directly affirm that their work does not report information on the metric quality of the model but only about texture characteristics [37]. However, in the paper by Webb et al. [36], the authors do report metric data because their aim was to evaluate cracking dimensions, and their progress over time, by use of infrared images.

The last relevant step in the procedure concerns the use of tri-dimensional UVF models. This may be expressed in the following question: how can we make these models available to the end users, such as conservators, restorers, art historians, archeologists, students, scholars, etc.?

In fact, all the examined papers in the literature only present 2D images extracted from 3D models. In light of this, we had the idea of using the Sketchfab online platform, which was specifically developed for uploading and sharing 3D models [85].

Sketchfab is a commercial platform that allows 3D models to be uploaded. They can then be made available through a specific link, so that the end user can access them from different kinds of device, such as a laptop, tablet or mobile. For example, a restorer can check the software while at work by simply accessing the model and exploring it (rotating, zooming). Other platforms can also be used to archive 3D models and, above all, to make them visible and usable; however, in none of the articles in which 3D UVF data are presented is there mention of use of any type of visualization system.

The advantage of Sketchfab is that models can be visualized, rotated, zoomed in, etc., by any user who has the link. Moreover, if authorized by the owner of the model, the user can also download the model.

Some examples are reported in Appendix A, in which a specific link to Sketchfab is shown for each model [86]. By accessing through the link, it is possible to observe the model, to rotate it, and to zoom in on specific zones to check fluorescence details. The models can also be downloaded if authorized by the authors. Recently, another paper presented tri-dimensional models in UV-induced fluorescence on the Sketchfab platform, along with a vast library of examples available online [43].

Lastly, the original models obtained through the photogrammetric software are available in our laboratories in the traditional formats (.obj., .ply, .3ds, .pdf and others) to enable further processing operations using 3D modeling and/or mapping software. Mapping is highly relevant in conservation. It is normally used by restorers to put into graphical form information about artworks and operations performed on them, such as mapping of the state of conservation, cracks and surface detachments, as well as cleaning, consolidation, inpainting, etc.

## 3. Conclusions

To conclude this paper, we list some key points summarizing the main findings of our research about usage of tri-dimensional ultraviolet-induced fluorescence imaging in cultural heritage:Our review of the literature revealed the presence of several papers underscoring the great utility of ultraviolet fluorescence photography as a diagnostic tool to support restoration, starting with the first published articles dating from about a century ago.The use of UVF imaging in 2D is widespread; however, the same imaging in 3D is much less frequently applied due to the difficulty of the procedure, especially in the phase of acquisition of the images necessary for creating the 3D model.An innovative approach aimed at simplifying acquisition was proposed some years ago. This approach is based on photogrammetry and SfM, but with the key modification that the camera and the UV sources are fixed while the object to be acquired is rotated on a turntable. Acquisition is possible thanks to the completely dark background required for UVF capture, as described in detail in Section 2 above.One relevant finding concerns visualization and use of the 3D UVF models. We found that output is generally reported by showing 2D images that are extrapolated from the 3D model workflow, or even single frames used for creating the digital model. The use of online platforms such as Sketchfab may provide help in this regard.Lastly, the original models can be used for other applications, such as mapping of the state of conservation in the case of restoration activities, which is a usual procedure in the documentation phases that precede practical operations carried out on artworks.

## Figures and Tables

**Figure 1 jimaging-11-00245-f001:**
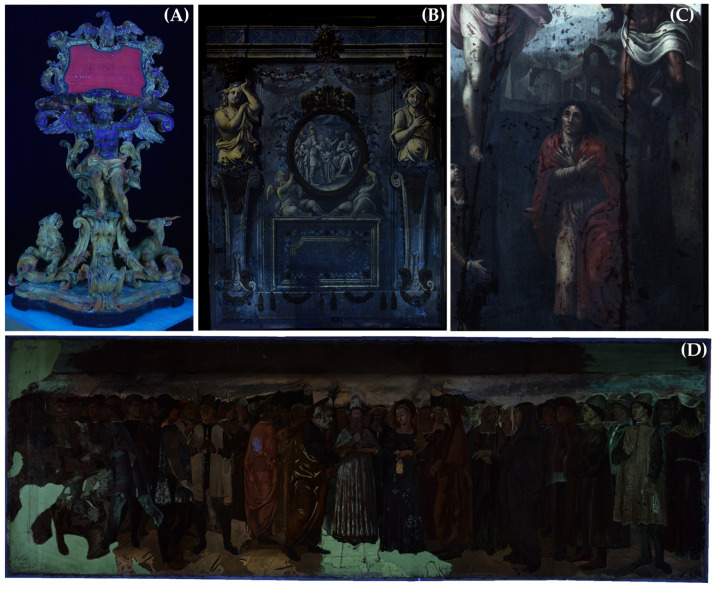
Some examples of UVF images from our laboratory archive demonstrating the utility of the UVF technique for the study of artworks, especially during restoration activities. (**A**) An 18th-century tetramorphic wooden lectern from the Church of the Certosa of San Martino (Naples, Italy), showing the progress of the cleaning operation (upper part cleaned, lower part uncleaned) in the Restoration Laboratory of the University of Tuscia; (**B**) the 18th-century wall painting in Palazzo Nuzzi at Orte (Central Italy), showing yellow fluorescence in the area of interest with inpainting with zinc white [21]; (**C**) detail of a 16th-century panel painting representing the Crucifixion and attributed to Michelangelo’s workshop, displayed in the Museum of Colle del Duomo in Viterbo, showing the presence of cracks, inpainting and grouting [22]; (**D**) a large 19th-century canvas representing the wall paintings of Cappella Mazzatosta by the Renaissance artist Lorenzo da Viterbo, showing an interesting use of two different kinds of white: yellow fluorescent zinc white, and lead white [12] (Reproduced with permission from L. Lanteri, C. Pelosi, Proceedings Volume 11784, Optics for Arts, Architecture, and Archaeology VIII; published by SPIE, 2021).

**Figure 2 jimaging-11-00245-f002:**
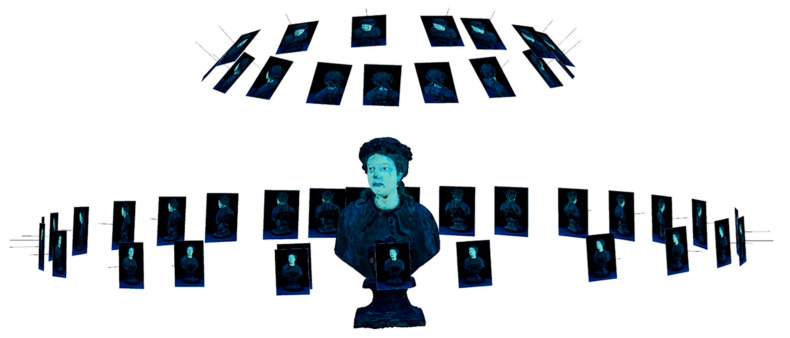
Creation of a tri-dimension UVF model starting from images acquired by rotating the object—in this case, a reliquary bust representing St. Rosalia—and keeping the camera and the UV sources fixed.

## Data Availability

All the 3D models are published on the Sketchfab platform.

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
