# Peer review of "Three-Dimensional Ultraviolet Fluorescence Imaging in Cultural Heritage: A Review of Applications in Multi-Material Artworks"

_2313-433X, 2025, doi:10.3390/jimaging11070245_

Round 1

Reviewer 1 Report (Previous Reviewer 2)

Comments and Suggestions for Authors

The authors have created a review paper. The topic is the application of ultraviolet fluorescence photography as a diagnostic tool to support restoration and this topic has been researched for about 100 years. This technique has only recently been applied to 3D shape analysis because there were problems that needed to be solved related to collecting data on site.

The solution was to photograph the 3D model in a darkened room, where the model was on a turntable on which it was rotated, and the camera and lights of the appropriate wavelength were fixed. Also, the background was black so that it could be easily removed during data processing. The technique used is Structure from Motion.

Such a model was exported to Sketchfab software, which is convenient for displaying 3D models to various users who do not necessarily have any special technical knowledge.

I suggest that the editors accept the work for publication. I believe that it will be interesting for a wide audience to read.

Author Response

Reviewer 1

The authors have created a review paper. The topic is the application of ultraviolet fluorescence photography as a diagnostic tool to support restoration and this topic has been researched for about 100 years. This technique has only recently been applied to 3D shape analysis because there were problems that needed to be solved related to collecting data on site.

The solution was to photograph the 3D model in a darkened room, where the model was on a turntable on which it was rotated, and the camera and lights of the appropriate wavelength were fixed. Also, the background was black so that it could be easily removed during data processing. The technique used is Structure from Motion.

Such a model was exported to Sketchfab software, which is convenient for displaying 3D models to various users who do not necessarily have any special technical knowledge.

I suggest that the editors accept the work for publication. I believe that it will be interesting for a wide audience to read.

Authors reply: I thank the reviewer very much for the precise summary of our article for the appreciation of the review

Reviewer 2 Report (New Reviewer)

Comments and Suggestions for Authors

This article is about 3D model calculated from UV induced fluorescence in visible range imaging. The title suggests it is a review of application for artwork analysis.

UVF in 3D is without doubt a powerful tool for cultural heritage community and it is true that very few papers are treating this subject.

But this article is difficult to follow as the structure of chapters are not clear. A lot of sentences are repeated. The chapter Discussion is not a discussion but already said remarks, as the conclusion. Two sentences talk about infrared imaging, that is not the subject.

The review miss technical informations :
- UVF is not explained at the beginning as UV induced fluorescence in the visible range;
- We don't know which range of UV illumination is considered. Emitting radiations change the responses. The tungsten-halogen lamps cited among others do not send UV. Artwork materials react to 365 nm +/- 10nm but also strongly to other UV radiations available in Led. 
- No indication about filters used to register images.
- No indication about poor UV sensitivity of new camera sensors.
- Line 116, intermediate focal lens is not 50mm for all camera formats.

If this article is a review of applications about artwork analysis, the minimum is to explicit main materials reactions under UV, that were extensively studied in 2D imaging or reflectance spectroscopy.

Other articles were already published by authors on this subject. The novelty highlighted by this article are self citations. 

It seems that novelty could be on the sketch fab visualisation tool. But unfortunately, this tool has been bought by a new company and will probably no longer exist. Museum scientists are actually searching for a new platform.

In conclusion, the purpose of this article is not clear, and could not be published without major revisions.

Author Response

This article is about 3D model calculated from UV induced fluorescence in visible range imaging. The title suggests it is a review of application for artwork analysis.

UVF in 3D is without doubt a powerful tool for cultural heritage community and it is true that very few papers are treating this subject.

Authors reply: we thank the reviewer for the comments and for having appreciated the subject of the paper.

But this article is difficult to follow as the structure of chapters are not clear. A lot of sentences are repeated. The chapter Discussion is not a discussion but already said remarks, as the conclusion. Two sentences talk about infrared imaging, that is not the subject.

Authors reply: the authors would like to thank the reviewer for the careful revision work and for the suggestions to improve the manuscript.

We tried to modify the papers in order to avoid repetitions. Discussion section has been completely removed, and some parts were moved in the previous paragraph in order to avoid repetitons.

The two sentences about NIR have been consequently removed.

Table 1 has been removed from the paper and inserted in a supplementary file.

Conclusions have been maintained as a synthesis of the paper with key points that, according to our opinion, are useful for the reader.

Anyway, some re-arrangements of the conclusions have been made

All changes and addition were made in red characters so they ae immediately visible

The review miss technical informations : - UVF is not explained at the beginning as UV induced fluorescence in the visible range;

Authors reply: we modified the text as suggested by the reviewer to better explain UVF as induced fluorescence in the visible range.

- We don't know which range of UV illumination is considered. Emitting radiations change the responses. The tungsten-halogen lamps cited among others do not send UV. Artwork materials react to 365 nm +/- 10nm but also strongly to other UV radiations available in Led.

Authors reply: some details about the UV lighting systems have been added in the paper.

- No indication about filters used to register images.

Authors reply: the filter typologies are indicated in the paper cited in the review. I don’t think that we have to report them in the article. The filter used in our laboratory, for example, are well-explained in the published papers and the spectra of the filters are reported as well.

- No indication about poor UV sensitivity of new camera sensors.

Authors reply: Commercial camera sensors certainly have a poor sensitivity to ultraviolet radiation, the fact remains that in the UVF technique the camera must record the effect induced by ultraviolet radiation on the painted surfaces which, as is known, is a fluorescence characterized by a lower frequency and a corresponding longer wavelength which is therefore recorded in the visible range. The camera normally used for UVF have good resolution.

- Line 116, intermediate focal lens is not 50mm for all camera formats.

Authors reply: thanks for the comments, we modified the sentence

If this article is a review of applications about artwork analysis, the minimum is to explicit main materials reactions under UV, that were extensively studied in 2D imaging or reflectance spectroscopy.

Authors reply: we have reported some examples in the introduction by showing UV induced fluorescence in the visible of artworks of different typologies in order to highlight the potentiality of this simple but useful tool widely used by restorers for their routine work.

Examples of fluorescence in pigments and varnished are reported as well as in restoration materials and inpainting.

Some further aspects are added in the paragraph 2.

Other articles were already published by authors on this subject. The novelty highlighted by this article are self citations. 

Authors reply: we think to have cited all published papers about tri-dimensional application of UVF technique. There are 85 references (now 86 because we added a further paper published on Sensors after our submission, so I couldn’t know of this paper!) in the manuscript only 11 by the authors, that is 13%: this is moderate self-citation, normally accepted without problems in all papers.

It seems that novelty could be on the sketch fab visualisation tool. But unfortunately, this tool has been bought by a new company and will probably no longer exist. Museum scientists are actually searching for a new platform.

Author reply: we have been using Sketchfab platform successfully for almost 10 years and without ever having had a problem and recently we renewed the subscription for a further year.

All our models, but also those of other institutions (see reference 43), are perfectly visible and can be also downloaded if authorized.

Sincerely, I don’t understand the problem.

If the problem of viewing and downloading 3D models from the platform were to materialize, similarly to other research centers, we will be able to migrate effectively to platforms that guarantee a comparable standard. I consider this observation pertinent but somewhat specious in a scientific context where those who claim to create 3D models in UVF, publish exclusively two-dimensional images and do not provide any access to three-dimensional models, except for the recent paper published on Sensor that initially we did not report in our review simply because it was not published at the moment of our submission.

In conclusion, the purpose of this article is not clear, and could not be published without major revisions.

This manuscript is a resubmission of an earlier submission. The following is a list of the peer review reports and author responses from that submission.

Round 1

Reviewer 1 Report

Comments and Suggestions for Authors

This manuscript reviews ultraviolet fluorescence (UVF) imaging in cultural heritage, focusing on the photogrammetric method for generating 3D digital models. It does not meet the necessary standards for publication in its present state. The paper exhibits limitations in its critical engagement with the broader literature, lacks a quantitative evaluation, and disproportionately emphasises the authors' prior work without a balanced scholarly analysis. The following points detail the factors that influenced the decision.

  • The paper focuses on outlining a methodology instead of engaging in critical analysis or synthesis of existing approaches. A review article must evaluate the comparative strengths, limitations, and impacts of techniques rather than simply presenting them. Assertions about the method's novelty and superiority are made without appropriate benchmarking or evidence-based support.
  • The proposed method lacks quantitative metrics, such as time efficiency, spatial resolution, and processing accuracy, to illustrate its improvements over existing techniques. The methodological section is deficient in detail regarding validation procedures, quality assurance, and the reproducibility of the created 3D models.
  • The authors characterise the paper as a review; however, it primarily serves as a summary of their previous research. A review must encapsulate a synthesis of developments across the field rather than function as a comprehensive project report. This paper lacks a novel conceptual framework or systematic comparison to substantiate its classification as a scholarly review.
  • The manuscript exhibits several grammatical errors, awkward phrasings, and excessively informal expressions that undermine clarity and the academic tone. Figures are accompanied by insufficient captions and contextual explanations; certain figures are labelled as “unpublished” yet lack adequate sourcing or clarification. Furthermore, Table 1 lacks clarity in its present format and fails to fulfil the review's essential purpose.
  • The discourse surrounding Sketchfab, although pragmatic, tends to overemphasize its advantages without adequately addressing the platform's limitations, such as long-term access, interoperability, data loss, and licensing issues. Alternative visualisation and archival solutions are minimally explored, and there is a lack of discussion regarding best practices in digital preservation standards.

Reviewer 2 Report

Comments and Suggestions for Authors

I commend the authors for their choice of topic and for the interesting way they present it.

I have some notes:

Figure 1 says "from our laboratory" but it would have been better to write name of  laboratory.

Line 164 “software ae” ?

Line 229  …NIR write in parentheses Near-Infrared
